# How Food Consumption Trends Change the Direction of Sheep Breeding in China

**DOI:** 10.3390/ani14213047

**Published:** 2024-10-22

**Authors:** Xiaoyu Wang, Wei Shen, Pan Wu, Chengli Wang, Jiahua Li, Di Wang, Wanfu Yue

**Affiliations:** College of Animal Science and Technology College of Animal Medicine, Zhejiang A&F University, Hangzhou 311300, China; wxyinhenan@163.com (X.W.); roy151918@163.com (W.S.); myhorse07@163.com (P.W.); wangchengli66619@163.com (C.W.); 19157737435@163.com (J.L.); 18642177530@163.com (D.W.)

**Keywords:** livestock industry, breeding direction, consumption demand

## Abstract

**Simple Summary:**

Along with the development of global consumer demand and rapid progress in science and technology, livestock products other than food have been largely replaced by non-livestock products, and the main role of the livestock industry has become the production of food, the structure of which has changed from diverse to monotonous. Traditional livestock breeds, such as sheep, which were once used to produce wool and skins, and cattle, which were used for producing arable land, are all bred for food production today. The expected market demand for specific livestock products directly influences the setting of breeding objectives. The rapid development of China’s overall economy has been accompanied by a rapid transformation of the local livestock industry, and many traditional sheep and goat breeds have been eliminated or selected to varying degrees because of their breed characteristics. As the top producer and consumer of mutton and goat meat, China’s history of the transition of traditional sheep breeds to specialized meat use is of high analytical value.

**Abstract:**

This paper discusses how food consumption trends have influenced the direction of sheep and goat breeding, particularly concerning the shift in sheep and goat farming and the development of the Hu sheep industry in China. Historically, sheep have been bred primarily for wool production, but with the increased demand for meat, the breeding direction has begun to shift toward meat use. As a major producer and consumer of mutton and goat meat, there is room for development of the meat sheep industry in China. In this paper, we review the development of China’s sheep breeding industry before and after the reform and the change in breeding objectives through examples, summarize the factors causing breeding changes in China’s sheep breeding industry, introduce the breeding achievements of the Hu sheep after its shift to meat use, analyze the reasons as to why this breed was selected by a vast number of farmers, and reveal the current decisive traits in the development of the meat sheep industry. This article also shows the improvement in Chinese people’s consumption abilities and the change from sheep breeding to consumption, providing China’s development experience as an example for today’s regions lagging in the global livestock industry. Breeding to respond positively to changes in the economy can better cater to the consumer food market.

## 1. Consumer-Led Shift in Sheep and Goat Farming

Sheep and goats, as two ancient breeds of livestock, have a long history of being farmed, providing meat, milk, wool, fleece, leather, and other products for human consumption. Prior to the 1950s, mutton and goat meat were by-products of the development of the wool industry, and the dominant position of sheep farming in agro-industrialization was due to the wool industry. As a textile material, wool was one of the first natural fibers used by mankind. In the 12th century, due to the large demand for wool, the Spaniards bred the world’s first wool sheep breed—Merino sheep—through crossbreeding and appearance selection. With the development of cotton and hemp and the chemical fiber textile industry in the 19th century, the wool market gradually weakened. At the same time, due to the modernization of the production model brought about by the increase in productivity and the gradual emergence of mutton and goat meat consumption, sheep breeding also changed. In the 20th century—around the 1930s—meat breeds became specialized for selection and breeding; goat breeds and sheep breeds such as Dorper sheep, Boer goats, Australian White sheep, White Suffolk sheep, and other world-renowned specialized meat sheep emerged in this historical age. By the 1960s, wool had completely lost its price advantage over synthetic fibers [1], and the decline of the wool industry as a pillar industry had a negative impact on the entire sheep industry.

Along with the leap in food consumption capacity in the international market in recent years, the consumption of all types of meat products has also increased. World/China per capita food and protein consumption data published by the Food and Agriculture Organization of the United Nations (FAO) and China’s National Bureau of Statistics (CNBS) show that, from 2010 to 2021 (Figure 1), world per capita food and protein consumption increased by 5.1% and 7.1%, respectively, and China’s per capita food and protein consumption increased by 11.5% and 14.5%, with the increase in protein consumption being greater than that of food consumption. The increase in protein consumption is also greater than that of grain consumption. Meat consumption in China has increased from only 18.2% in 2014 to 25.3% in 2022. Mutton and goat meat, with its low-fat and high nutritional advantages, has been able to increase its share in traditional meat production and gradually replace wool and sheepskins as the centerpiece of the traditional sheep farming industry. The economic benefits of mutton and goat meat have also surpassed those of wool and sheepskins, and, with the exception of a few developed countries with established wool industries that still hold a dominant market position in the wool industry, the world’s sheep industry is generally moving from wool to meat.

## 2. Transformation of China’s Sheep and Goat Farming Industry

Because of the genetic and biological similarities between goats and sheep and their similar economic and production purposes in Chinese history, the two breeds are symbolized by the same Chinese character “Yang” and are also counted together for production. China’s sustained economic growth, rising incomes, and upgraded consumption structure have shifted the demand for agricultural products from food and vegetables to diversification. According to “China’s agricultural outlook report (2023–2032)” [2], the growth in food demand mainly comes from the faster growth in food consumption; food consumption is expected to increase by 13.4%, while per capita ration consumption is declining, in line with the changes in the food consumption structure. At present, between China’s domestic and foreign types of meat products, price differences still exist; however, with the further opening up of the domestic market and increased domestic demand for meat products, meat imports have increased, and domestic meat products’ self-sufficiency rate is not accompanied by development and improvement in internal productivity.

China was the first major producer and consumer of mutton and goat meat [3], and its germplasm resources are very rich. Mutton and goat meat production in 2022 accounted for 31.1% of the world’s output of mutton and goat meat, but, at present, there is still a gap between it and the levels produced by developed countries, and specialized hybrid meat parent–seed sources that are suitable for large-scale factory food production still need to be imported from abroad; the specialization of meat parent varieties are very scarce.

### 2.1. Development of Sheep and Goat Farming in China

The self-sufficiency rate (SSR) of livestock and poultry products was calculated using a formula put forward by Wang Jating [4]. This formula utilizes China’s mutton and goat meat production (P), imports (I), exports (E), and stock increase (S). The data used for this calculation were sourced from the global food balance sheets, 2010–2021, published by the Food and Agriculture Organization (FAO) and the China National Bureau of Statistics (CNBS).
SSR = P/(P + I − E − S) × 100%(1)

The self-sufficiency rate of mutton and goat meat products in China from 2010 to 2021 was calculated. This formula analyses the self-sufficiency of major domestic livestock and poultry products since the reform and opening up and forecasts the development trend, providing scientific support for the supply of livestock and poultry products in China during the Fourteenth Five-Year Plan period.

In summary, China’s mutton and goat meat production and import quantity have experienced significant growth from 2010 to 2021, increasing by 28.72% and 323.08%, respectively (Table 1). This growth has been accompanied by an increase in the proportion of mutton and goat meat production in overall meat production, as well as a decrease in export quantity and inventory. This indicates that China’s internal consumer market has a growing demand for mutton and goat meat products. However, China’s self-sufficiency rate for mutton and goat meat is decreasing, indicating that the country’s lamb production capacity has not kept pace with its consumption capacity and that there is still a gap between the production level of China’s sheep farming industry and that of advanced agricultural countries. Stable returns can ensure the development of the industry, and the future market demand determines the breeding objectives of the breeding staff. After 15–16 years of decline, the production price index of live sheep continued to grow, proving that China’s sheep industry has full potential for development.

### 2.2. Distribution of Mutton and Goat Meat Industry in China

Currently, China (Mainland) is divided into four main meat-sheep-advantageous areas, which are the Central Plains, Middle East, Northwest, and Southwest [5]. Figure 2 illustrates that the Inner Mongolia and Xinjiang provinces are the largest meat sheep production areas in China (Mainland). However, the production of mutton and goat meat in traditional regions still has a fairly dominant role. Due to climatic differences, the northern breeds of meat sheep and goats cannot adapt to the production environment of the south. Therefore, the development of local sheep and goat breeds in the south, supported by the Chinese Government’s north–south migration policy, has become a current breeding trend.

## 3. Several Direction Shifts in Chinese Sheep Breeding

### 3.1. Before “Reform and Opening Up”

Since the creation of the People’s Republic of China, China’s sheep and goat industry has expanded at an unprecedented rate. Initially, China’s sheep population consisted of primitive local breeds such as the Mongolian sheep, Tibetan sheep, Hetianyu, Kazakh sheep, etc. With a population of 28.52 million wool-producing sheep reported in 1950, however, only 30,000 of these were fine-wool breeds. Meanwhile, the goat population at this time was entirely made up of Chinese native common goats.

Over the course of the next thirty years, China’s sheep and goat industry experienced significant growth. By 1980, the country’s sheep population had increased to 106.63 million, a 400.56% increase from 1949. This significant growth was primarily due to the introduction of foreign breeds, with the proportion of local breeds decreasing to 43.30%. Similarly, the goat population also experienced significant growth. By 1980, the country’s goat population had increased to 80.68 million, a 500.19% increase from 1949. However, unlike the sheep population, the proportion of local breeds in the goat population remained relatively stable, decreasing only slightly to 77.06% [6]. Overall, the growth of China’s sheep and goat industry since the founding of the People’s Republic of China has been remarkable, with significant changes in the types of breeds used and the proportion of local breeds.

Not only did the number of sheep and goats experience great growth but also the quality of sheep and goat farming products, which saw significant improvement. Due to the demand for wool, China’s sheep and goat breeding was mainly based on wool/meat from fine-wool sheep and semi-fine-wool sheep; based on the work at that stage, a total of 22 breeds of fine-wool and semi-fine-wool sheep were used in production.

### 3.2. After “Reform and Opening Up”

Traditional sheep farming is a more popular industry compared to pig farming and cattle farming due to its low investment, rapid results, and high profits. This industry is also accessible to all, including the elderly and vulnerable populations, which may help improve rural labor conditions. Many traditional Chinese sheep and goat breeds, such as Black Goats, beach goats, Tibetan sheep, and alpine fine-wooled sheep, thrive in areas with poor ecological conditions, which are often associated with poverty. These breeds have long been a source of livelihood for farmers in these areas. Therefore, the rapid development of the sheep industry is closely related to the government’s poverty alleviation policies. The family-farm-based meat sheep industry, which is promoted and supported by various poverty alleviation programs, has played a significant role in poverty alleviation, increasing employment opportunities and enriching the rural economy. The rapid development of the meat sheep industry is a testament to the government’s effective poverty alleviation work.

After China’s reform and opening up, remarkable achievements were made in the promotion and utilization of sheep breeds for meat. Breeds such as the Dupo, Boer goat, Small-tailed Han Sheep, and Black Goat have been widely introduced and bred for their unique advantages. These breeds have significantly contributed to the development of China’s animal husbandry industry.

Nevertheless, these breeds encountered distinct limitations during the promotional and utilization phase, mirroring the changing production environment. As China’s rural reform continues, the northern region has banned grazing, re-environmentalizing ploughland and transforming it into forests and grasses. Additionally, the grassy hills and slopes in the southern region have been closed to forests, resulting in a drastic reduction in the available grazing area.

Only breeds that can adapt to barn-feeding production will be able to cope with the wave of industrialization of the farming industry under the policy of returning pasture to grassland. In the following section, we will introduce several large-scale and representative breeding achievements in the history of sheep and goat breeding after reform and opening up, and we will analyze the strengths and weaknesses of their breeds to reflect the changes in China’s meat sheep breeding industry and the reasons behind them. The excellent genetic resources of the Hu sheep as a meat sheep are further described, and arguments are presented for the reasons why this breed is widely selected at present.

#### 3.2.1. Dorper Sheep

Dorper sheep, native to South Africa, were imported from Australia to China in 2001 for the purpose of meat production. The breed has proven to adapt well to heat, drought, cold, and other climatic conditions, thereby becoming the preferred parent breed for crossbreeding and for the development of faster-growing, higher-yield meat breeds. The breed has been used extensively as the male parent in breeding programs, resulting in significant improvements in meat production and quality [7].

Dorper sheep are renowned for their resilience and adaptability to roughage-based diets, but when compared to breeds such as Small-tailed Han Sheep, Hu sheep, and others, they still exhibit inferior performance. In order to optimize their growth potential, Dorper sheep require a balanced diet that incorporates a certain amount of concentrate feed. Moreover, they are not well suited to hot and humid environments and are prone to liver flake schistosomiasis and coccidiosis in lambs. This makes them less suitable for the climate of southern China. Dorper sheep also have difficulty adapting to confinement feeding, with each worm species showing a higher infection rate than in Hu sheep flocks [8]. Despite the significant contributions Dorper sheep have made to the global meat sheep industry and their long-standing role as excellent male parents in China’s meat sheep breeding work, they have demonstrated poor disease resistance and high nutritional requirements, as well as high management requirements, making it challenging to effectively integrate them into family farm production models.

#### 3.2.2. Boer Goats

Boer goats are highly favored for their robust build, swift growth trajectory, succulent meat, and a myriad of other virtues. These exceptional ruminants were first imported from Europe, specifically Germany, to China in 1995. The introduction of Boer goats to China necessitated significant investment from the government, industry, and private entities in the promotion, cultivation, and utilization of germplasm resources.

Despite the remarkable qualities of Boer goats, they also have clear challenges that must be overcome. Firstly, the high price of purebred Boer goats and their heightened nutritional needs are unfavorable to Chinese goat farms, most of which operate as family farms. This creates a challenge when promoting the breed. Secondly, Boer goats have subpar reproductive performance, characterized by late sexual maturity, a marked seasonality of estrus that peaks in autumn, and subtle estrus signs that typically only include tail wagging and infrequent chirping. This makes it challenging to accurately time mating, requiring farmers to have a high level of technical expertise and extensive experience in rearing management [9]. Additionally, Boer goats have limited adaptability, particularly in cold regions where their cold tolerance is poor, limiting their breeding in key breeding areas in China, such as the northeast and northwest. Despite the government and enterprises investing in and promoting Boer goats due to their exceptional meat production performance, the high price of Boer goat breeding and the higher difficulty of rearing make it challenging for ordinary farmers to adopt the breed.

#### 3.2.3. Small-Tailed Han Sheep

The Small-tailed Han Sheep, a native breed in China, is classified as a national second-grade protected animal. It possesses extraordinary characteristics including a large body size, rapid growth and development, early sexual maturity, year-round estrus, multiple lambs, tolerance to rough feeding, and stable genetic performance. The production model mainly includes confinement feeding supplemented by grazing. As a local breed in China, the unique breed advantage of the Small-tailed Han Sheep has made it an ideal female parent for meat sheep breeding from the beginning.

Nationwide promotion of the Small-tailed Han Sheep began in the late 1980s. In this time period, backing from local government and related organizations was instrumental in promoting Small-tailed Han Sheep breeding [10]. The goal was to enhance the growth and development of the meat sheep industry through scientific introduction and breeding methods. Today, the Small-tailed Han Sheep has seen great success in meat breeding, leading to remarkable improvements and a variety of meat traits. In the early 1970s, the number of modified breeds was less than 30,000, but by 2006, this number had grown to 4 million, with a wide distribution across various regions of China.

In summary, the Small-tailed Han Sheep possess a number of desirable traits, such as early sexual maturity and high fertility. However, they also have their share of disadvantages. Despite being well suited for meat production, their low feed conversion ratio and underdeveloped body conformation limit their potential. In addition, the breed is prone to many diseases, including brucellosis, toxemia, anthrax, and foot-and-mouth disease, as well as diarrhea and parasitism, due to food and water contamination during rearing [11]. This necessitates preventive measures, including regular gastric lavage, which further increases the breeding cost. These factors contribute to the challenge of advancing the breeding of Small-tailed Han Sheep in the Chinese meat sheep industry.

### 3.3. Hu Sheep

As animal husbandry, the ardor for animal production, and the massive adoption of modern livestock breeding theories and methods are exponentially seen, they have catapulted animal production to levels not seen before. However, this enormous progress comes with a price, and that price is an enormous threat to the genetic diversity of traditional animal resources across the nation. The genetic diversity of the Chinese livestock resources is under serious threat. According to a second national survey of livestock and poultry genetic resources conducted between 2006 and 2012, there has been a 279% decline in indigenous breeds, accounting for 54% of all breeds. This is a drastic increase from 42% in 2004 [12]. The survey also found that 15 indigenous breeds have become extinct, 55 are endangered, and 22 are on the brink of extinction. This means that 14% of the total animal species are at risk of extinction or are already on the brink of extinction.

The following expounds on the justifications for the important role of the genetic resources of the Hu sheep, a time-honored Chinese breed, in today’s centralized animal husbandry industry.

#### 3.3.1. Origin of Hu Sheep

According to the National Catalogue of Livestock and Poultry Genetic Resources published by the Office of the National Livestock and Poultry Genetic Resources Commission of China in 2020, China possesses a total of 104 local sheep and goat breeds. This vast array of breeds, which includes 44 sheep and 60 goats, is the result of both long-term natural selection and artificial selection. These distinctive breeds have evolved in response to the changing needs and preferences of the Chinese people, resulting in a wide range of genetic diversity and unique breed characteristics [13]. These local breeds are the backbone of sheep and goat farming in various regions and are also the primary source of mutton and goat meat in China.

The sheep breeds found throughout China come from the three dominant systems of Mongolian, Kazakh, and Tibetan sheep. Within these systems, there are six economic types: coarse-wooled sheep, meat-fat sheep, lamb-skinned sheep, fur-skinned sheep, fine-wooled sheep, and semi-fine-wooled sheep. Hu sheep, in particular, have long been reared as lamb-skinned sheep and are also the only wool and meat sheep breed in southern China.

#### 3.3.2. Confinement Feeding

At present, China’s meat sheep and goat breeding operation is primarily operated by family-owned businesses, showing a low degree of organization, dispersion, lack of regulation, poor standardization, and intensification, as well as limited breed specialization in confinement feeding. Traditional grazing sheep and goat breeds, which were previously used for fur and wool production, are not suitable for large-scale intensive and large-scale farming due to various issues, including low lambing numbers, low lactation performance, low lamb survival rates, small litter weight, seasonal turnout, and long production cycles, which ultimately leads to a lack of economic benefits when transitioning to meat production [14].

The confinement feeding mode has proven to be effective in improving production performance and economic efficiency; it also reduces the reliance of sheep and goat farming on the production environment, propelling the meat sheep and goat industry towards a modernized commercial mode of production. Research studies have indicated that sheep and goats reared under confinement feeding regimes tend to exhibit superior lambing rates, weight gain, slaughter rates, and other production performance metrics [15].

Following China’s strategic policy of “returning ploughland to forests and forbidding grazing”, a significant portion of pastures have been closed for grazing. As such, the confinement feeding method has become an actionable trend that is replacing grazing [16]. The consistent long-term barn-feeding tradition of selection and breeding has paved the way for Hu sheep to evolve into a specialized barn-feeding breed. Hu sheep possess early sexual maturity, can mate and lamb in the same year of birth, and, due to their four-season estrus characteristics, can undergo year-round breeding and multiple fertility. Hu sheep are characterized by their lack of horns, timid and gentle temperament, diverse food preferences, ease of management, and their ability to adapt to high-density breeding modes without the need for specialized sports ground breeding. Therefore, Hu sheep are recognized as the key to industrializing the meat sheep industry in China.

#### 3.3.3. Reproductive Performance

The Hu sheep reaches sexual maturity at a relatively early age, typically at 6 months old. This early maturity allows them to become first-time breeders. Similar to other sheep, Hu sheep experience seasonal heat cycles, with rams typically reaching sexual maturity at 8–10 months old, while ewes can breed as early as 6–8 months. In terms of estrus cycles, the average duration for adult Hu sheep is approximately 17 days, with a gestation period typically lasting about 148 days.

The exceptional reproductive capabilities of the Hu sheep are attributed to its high level of multiparity. According to this group, the lambing rate of unselected Hu sheep primiparous ewes reached an impressive 163.2%. By comparison, the lambing rate of Hu sheep through ewes is even more impressive, at 235.7%. After years of selective breeding, the lambing rate of Hu sheep has increased to an astounding 218.2%, with the lambing rate of ewes reaching 240.9%. The newborn weight of lambs was 3.1 kg for males and 2.9 kg for females, with the weaning weight of lambs at 45 days of age being 15.4 kg for males and 14.7 kg for females. The weaning survival rate of lambs was an impressive 96.9%.

The FecB gene is a major effector gene to increase the fertility of ewes, which can increase the follicle size, increase the number of ovulations, increase the level of reproductive hormones in ewes, and control the expression of reproduction-related genes to improve the fertility of ewes. The FecB gene is widely found in Hu sheep populations and conforms to the Mendelian law of inheritance [17]. Although the gene is dominant and able to increase the body weight of sheep at 90 days of age [18], the gene has some negative effects on the early growth and development of Hu sheep [19]; therefore, the early nutrition and post-weaning compensatory nutrition of Hu sheep resulting from multiple births are very important, as is the management of separate flocks, ensuring the feeding ability of the lambs, and reducing the negative impacts of weaning and barn change stress—all measures to improve the rate of growth and the development of lambs.

#### 3.3.4. Excellent Maternity

The correlation between lactation and the lambing number of sheep is generally positive [20]. As a multi-lambing breed, lambing ewes of the Hu sheep must have the ability to suckle lambs from multiple births, and the output of a lactation period (120d) of Hu sheep is more than 100 kg, which can be up to 230 kg through supplemental concentrates [21]. The high milk production of Hu sheep ewes means that if lactation is not fully consumed after birth, it will accumulate and cause udder swelling. Therefore, lactating ewes are also allowed to suckle non-biological lambs, making it easier to manage the lactation period of Hu sheep lambs and eliminating the need for separate pens before weaning. The high survival rate and early growth rate of Hu sheep lambs are attributed to the strong lactation performance of Hu sheep ewes.

#### 3.3.5. Environmental Adaptability

The Hu sheep is remarkably versatile, thoroughly adapting to various climatic conditions from its spread throughout China. A combination of long confinement feeding conditions and suitable feeding methods have allowed the Hu sheep to excel in hot and humid climates in the south and dry and cold climates in the north. One notable feature is the Hu sheep’s unique adaptation to the dry climate of Gansu due to its ability to directly transition from the humid climate of Zhejiang. The Hu sheep’s diet is incredibly diverse, as it eagerly accepts grasses, wild weeds, silage, crop stalks, mulberry leaves, fruit tree leaves, dregs of medicines, bamboo shoots shells, and more. This adaptability enables them to forage from a wide range of sources, making it easy to purchase feedstuffs. Additionally, the Hu sheep is the only sheep breed in the world that feeds at night, with night feeding accounting for more than 50% of the total feeding. Therefore, the Hu sheep has significant potential for factory and large-scale breeding in various climatic environments.

#### 3.3.6. Meat Quality

Hu sheep muscles are nutrient-rich, complete with a range of essential amino acids, lysine, leucine, valine threonine, and additional amino acids. Lysine content constitutes 31.7% of the total essential amino acids. The content of other nutrients, except water, amplifies with age, and the crude protein content in the muscles of adult Hu sheep reaches 22.6%. Under identical feeding management conditions, the nutritional characteristics and muscle fiber tissue characteristics of Hu sheep muscles are influenced by age, size, and other factors. The muscle fiber diameter and fiber density of muscles of different ages and different parts of the muscle are distinct, with weaned lamb meat demonstrating the most tender muscle [22].

#### 3.3.7. In Line with the Development Trend

In the late 1970s, the stock of Hu sheep reached a peak of 2,540,000 animals. Since then, the number of these sheep has declined rapidly due to changes in agricultural production patterns and market demand as a result of China’s rural reforms. By the mid-1990s, the focus of Hu sheep production shifted from primarily producing hides to primarily producing meat. Now in the 21st century, the number of Hu sheep is steadily increasing; by the end of 2006, there were 1,127,000 heads in stock, with 92.65% of them located in Zhejiang province and 7.35% in Jiangsu province. The scale of rearing Hu sheep is expanding, and the rate of slaughter continues to rise. In recent years, China’s Hu sheep meat production has been stable and continuously growing, from 1.98% of the overall production of mutton and goat meat in 2015 to 2.80%, with a high growth rate of 5.5% to 9.5% per year (Figure 3). The main production bases of Hu sheep industry are located in Zhejiang, Jiangsu, Shanghai, and other provinces that are vulnerable to the overall production of mutton and goat meat; therefore, the development and consolidation of the Hu sheep farming industry is also an important symbol of the current migration of China’s animal husbandry industry to the south.

The rapid development of Hu sheep is the result of a combination of factors, including the inherent strength of the species, government support, the application of new breeding methods, and increasing demand from the market. All these elements intertwine to form the distinctive history of Hu sheep breeding and contribute to the effective development of the species’ genetic resources.

## 4. Breeding Achievements of Hu Sheep for Meat Use

Before the 1960s, small-scale operations as the mode of production of Hu sheep breeding was mainly a spontaneous work of farmers due to the lack of scientific guidance on breeding. Farmers mainly considered whether Hu sheep were economically viable, while not considering the need for the protection and retention of the species, leading to the loss of a substantial portion of the purebred bloodline of Hu sheep.

The Institute of Animal Husbandry of the Chinese Academy of Agricultural Sciences recognized the potential of Hu sheep genetic resources and the importance of purebred breeding work and began the selection of this breed. Their efforts have helped to retain Hu sheep germplasm resources. However, the breeding objective at that time was mainly to transform lamb skin to achieve the purpose of creating more foreign exchange, which led to the culling of many Hu sheep with excellent meat production potential. The lack of organized and planned selection and breeding of Hu sheep also contributed to the existence of large differences in Hu sheep germplasm in different regions, resulting in the wool and meat direction of the average production level being mediocre.

In recent years, the rising maturity of Hu sheep meat breeding, increased scientific research input, and enhanced internal industrial collaborations have resulted in notable enhancements to the various meat qualities of Hu sheep. These advancements have led to a significant improvement in the performance of Hu sheep meat, resulting in notable scientific research outcomes and, consequently, a significant enhancement in the economic profitability of Hu sheep meat.

### 4.1. Slaughter Month and Weight of Rams

The overall purpose of meat sheep and goat breeding is to obtain the highest quantity and quality of mutton and goat meat at the lowest production cost; the quantity and quality of sheep are usually expressed in terms of carcass weight and carcass quality, of which the carcass weight is mainly dependent on the slaughter weight (end of fattening weight). Hu sheep experience early growth and fast development under normal feeding conditions; rams at 6 months old can reach more than 70% of the body weight of year-old sheep. The proportion of net meat rams is greater than the ewes; at this time, slaughter can achieve maximum economic benefits. A 1-year-old Hu sheep reaches more than 90% of the body weight of the adult sheep. Therefore, in this paper, the evolution of slaughter month and slaughter weight of Hu sheep in ram fattening was selected to demonstrate the breeding results of Hu meat sheep.

As shown in Table 2, the age at slaughter changed from October to the present six-to-eight months, and the average carcass weight per ram increased by 29.62 kg from common Hu sheep to selected Hu sheep, nearly doubling the meat production capacity. And the selected core Hu sheep breeding population’s meat production capacity in the industry’s leading areas such as Huzhou and Hangzhou is significantly better than in ordinary Hu sheep; advanced core breeding results regarding Hu sheep meat production capacity can be effectively improved after the incorporation of areas lagging behind in this capacity.

### 4.2. Lambing Rate

The importance of the lambing rate in the meat sheep industry should not be overlooked, as it is directly related to the economic benefits and development potential of sheep farming. A high lambing rate means that the culling rate can be increased, and the production cycle and generation interval of the flock can be shortened, thus increasing the meat production capacity and accelerating the breeding work.

The reproductive characteristics of Hu sheep, such as four-season rutting and high fertility, make them highly beneficial to the economy and development prospects of the breeding industry. As shown in Table 3, the purebred reproduction rate of Hu sheep increased from 214.1 percent to 253.3 percent from 1986 to 2023 and stabilized. Purebred breeding of Hu sheep in different production environments in various regions can achieve better results than for all types of crossbreeds. The high reproductive rate of Hu sheep, combined with artificial insemination, can make full use of the advantage of multiple births in Hu sheep and reduce the breeding cycle of new lines of Hu sheep for meat.

### 4.3. Breeding Work for Hu Sheep

Breeding sheep for meat originated in the late 18th century in the UK with the Southdown and Leicester sheep. In the mid-to-late 19th century, countries around the world successively introduced meat and wool breeds from the UK to crossbreed with local Merino sheep or other local sheep to breed new meat or wool breeds. Li Fadi divided today’s meat sheep breeding into two technical routes; one is the use of the mid-to-late nineteenth-century-originated Nanchu sheep and Leicester sheep to form meat and wool varieties and local breeds for crossbreeding new meat sheep breeds. Countries based on their resource allocation characteristics choose a widely distributed and well-adapted local breed as the female parent to establish an economic hybridization model and achieve better efficiency of meat sheep production. The other is the use of original breeds, continuously selected and bred to improve, purify, or select new types [34]. At present, the breeding of Hu sheep mainly continues in the second technical route mentioned above.

As China’s excellent local breed, Hu sheep have been widely utilized throughout the country; with the implementation of the project of “retreating from pasture to grass” in recent years, Hu sheep are suitable for confinement feeding and are also widely used for breeding in poverty alleviation work. At present, Hu sheep breeding farms in Zhejiang province, especially the core breeding plant concerned with the selection of the breed, can breed sheep in bulk for the country, and even the world, providing high-quality breeding sheep.

In addition to the core breeding plant of this breed that insists on developing the production potential of Hu sheep itself, there are also peripheral breeding plants that try to provide other sheep breeds with excellent meat production performance, such as Dorper sheep [7], Nanchu sheep [35], etc. to improve the various types of meat-producing traits of Hu sheep. Most of the regions of the country have attached importance to the multi-lambing trait of Hu sheep by crossbreeding local traditional rams with Hu sheep ewes to obtain the multi-lambing and good adaptation traits of Hu sheep that are suitable for local consumption habits and growth. These traits are suitable for local consumption habits and rapid growth and development and result in high slaughtering performance and relatively good meat quality in new meat sheep lines [36]. Although the introduction of other breeds may briefly improve various traits of Hu sheep, the establishment of hybrid advantages is based on pure parent breeds to ensure genetic gain based on additional enhancement [37] and should not be sacrificed at the expense of the genetic improvement of the pure breed. Therefore, breeding work to develop the potential of Hu sheep by breed selection is the ballast of the breeding work of Hu sheep [38].

In terms of breeding technology, the development of Hu sheep breeding technology simultaneously uses methods such as body appearance selection, individual phenotypic value selection, breeding value selection, molecular marker-assisted selection, and genomic selection [39]. With the rapid change in new technologies such as the Internet, big data, and artificial intelligence, Hu sheep breeding technology has entered a new stage. In particular, the accuracy and efficiency of performance measurement and the ability of histological big data processing has been rapidly developed; computed tomography, sheep face recognition, machine learning, multi-omics association, gene editing, Internet of Things, artificial intelligence, and other technologies have been applied to the field of meat sheep breeding, promoting innovation and upgrading breeding technology for Hu sheep with high throughput, high efficiency, and high accuracy [40]. Breeders have been encouraged by various projects to introduce advanced technologies into the front lines of breeding Hu sheep.

The sheep genome was successfully assembled after 2010, leading to the popularization of commercial single nucleotide polymorphism (SNP) microarrays throughout countries with advanced sheep breeding industries. Large reference populations for genomic selection were established and implemented sequentially in subsequent years. In 2024, Wang Weimin successfully developed a high-performance liquid SNP microarray encompassing 45,052 representative, global, and functional loci [41]. This was an important development for the advancement of global sheep genome selection breeding technology, and today, this technology has already been successfully implemented in the Hu sheep industry.

### 4.4. Hu Sheep Breeding Associations

Breeding associations play a vital role in organizing sheep and goat breeding systems in areas with well-developed sheep and goat breeding industries. They operate in a market-oriented manner, providing organizational safeguards for ongoing breeding and selection of breeds to maintain a competitive advantage. Family sheep farms, as the primary production and operation units, are the fundamental guarantee for the successful execution and implementation of the breeding plan.

At present, the Hu sheep industry does not have the same credibility as the pigs, cattle, and chickens, which have formed their high-credibility organizations. Although the government in Zhejiang organizes annual competitions and exchanges for Hu sheep genetic resources, the industry’s slow establishment of influential associations has created a gap in the evaluation system that can accurately reflect the professional standards of Hu sheep farmers. This has led buyers to have difficulty making practical comparisons when purchasing Hu sheep, while the lack of healthy competition among farmers is impeding further growth of Hu sheep breeding.

## 5. Conclusions

Compared to other traditional breeding industries such as pig and cattle breeding, the sheep and goat breeding industry in China shows notable features such as low investment, quick turnover, and high efficiency. Large-scale production in family farms for sheep and goats does not necessitate the use of complex breeding technology, making it an ideal industry for areas with relatively weak economic endowments. Currently, with the support of international organizations or non-governmental organizations in developing countries, local farmers have received training and technical support, enhancing their skills and knowledge of sheep and goat breeding for meat. Advanced sheep and goat breeding technologies and management experiences have been rapidly introduced to economically underprivileged regions around the world, generating foreign exchange for these economies, providing employment opportunities, and promoting economic development and the enrichment of the economic structure. The breeding transformation path of sheep and goats for the meat industry in China provides valuable experience for the development of the livestock economy in developing countries.

The secondary development of Hu sheep underscores the fact that livestock and poultry resources, as integral components of biodiversity, serve as the primary material foundation of human survival and development, alongside being a strategic resource that is non-renewable and capable of addressing unforeseen breeding needs in the future. Throughout the global market’s expansion, vast germplasm resources have not been fully utilized or effectively developed, leading to their potential extinction due to their inability to generate substantial economic benefits compared with more dominant varieties, and the industry has increasingly gravitated towards monospecificity. However, a single specialized livestock product cannot cater to the diverse dietary preferences of various regions. It is thus imperative that we actively protect and develop the existing livestock and poultry germplasm resources to meet the escalating diversified needs of consumers.

Our review has uncovered how food consumption patterns are continually evolving, which directly influences the planning and direction of all livestock breeders. Breeders who can adapt quickly and adeptly to economic shifts stand to capture the initial advantage in the food consumption market.

## Figures and Tables

**Figure 1 animals-14-03047-f001:**
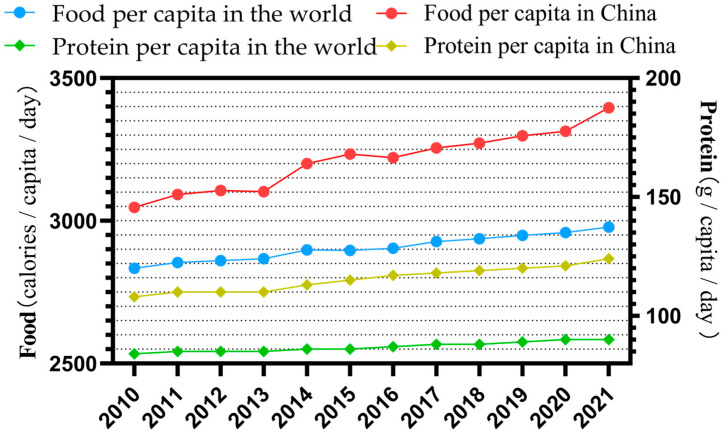
Per capita food and protein consumption in the world and China. (Data sourced from FAO and CNBS).

**Figure 2 animals-14-03047-f002:**
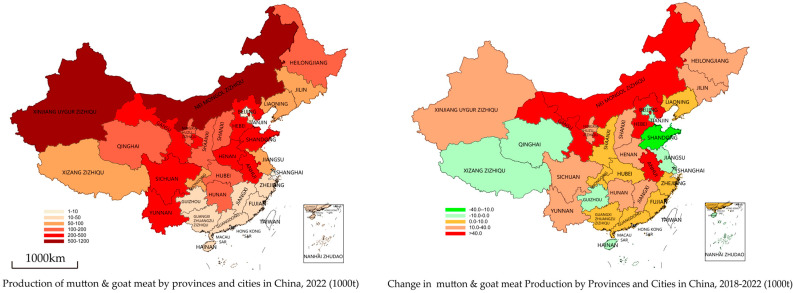
Production of mutton and goat meat and its changes in the provinces of China (Mainland).

**Figure 3 animals-14-03047-f003:**
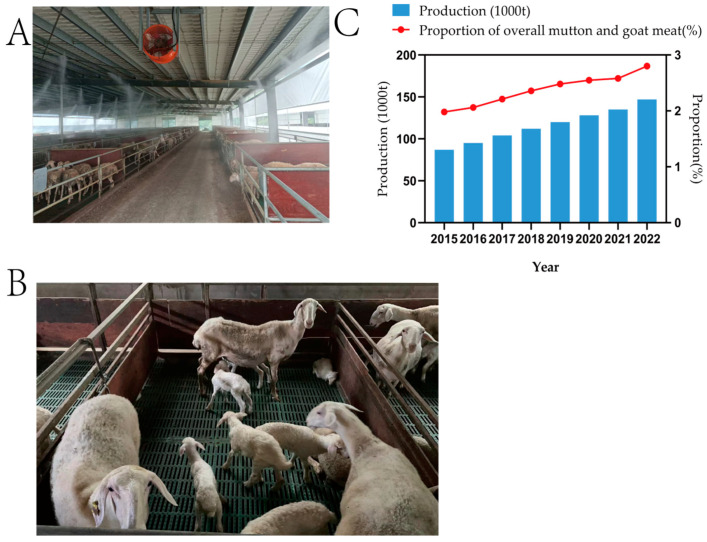
(**A**) The production environment of Hu sheep confinement feeding, Hu sheep can accept a full confinement feeding mode of production, therefore requiring less labor, and can be produced on a large scale in non-plain areas. (**B**) Hu sheep are timid and docile and will suckle non-biological lambs; therefore, they can be managed in a uniform and efficient manner, with five ewes and ten-to-fifteen unweaned lambs usually housed in a single barn. (**C**) Breeds of meat sheep bred in countries with developed animal husbandry traditions are leading China, but the stock of Hu sheep, which is an indigenous breed, has been able to increase rapidly throughout the country because of the irreplaceable genetic resources of Hu sheep.

**Table 1 animals-14-03047-t001:** Indicators of various types of mutton and goat meat in China from 2010 to 2021. (Data sourced from FAO and CNBS).

Year	Production (1000 t)	Imports (1000 t)	Exports (1000 t)	Change in Stocks (1000 t)	SSR (%)	Proportion of Overall Meat Production (%)	Producer Price Index (%)
2010	4063	104	20	448	109.84	5.05	108.7
2011	3982	125	9	210	102.42	4.93	115.7
2012	4047	159	6	64	97.85	4.79	107.8
2013	4101	296	4	−8	93.18	4.75	109.1
2014	4278	327	5	63	94.29	4.89	100.8
2015	4401	252	6	93	96.64	5.05	89.4
2016	4605	245	5	100	97.05	5.35	93.6
2017	4713	279	6	103	96.52	5.46	107.1
2018	4753	351	5	109	95.25	5.42	114.7
2019	4838	420	3	110	94.03	6.26	114.3
2020	5015	392	2	111	94.73	6.29	110.4
2021	5230	440	2	117	94.22	5.68	102.3

**Table 2 animals-14-03047-t002:** Changes in slaughter month and weight of rams.

Year	Sample Size	Area (City)	Type	Slaughter Weight	Slaughter Month	Data Sources
1980	-	Hu Zhou	common sheep	35.69	10	Lin Jia [23]
2007	-	Hu Zhou	common sheep	45.25 ± 3.58	10	Lin Jia [23]
2009	56	Hang Zhou	common sheep	36.28 ± 4.64	6	Zhang Gaozhen [24]
2014	8	Hu Zhou	common sheep	36.77 ± 3.96	6	Lin Cangjun
2014	8	Hu Zhou	Selected sheep	39.68 ± 4.46	6	Lin Cangjun [25]
2019	8	Hu Zhou	common sheep	40.76 ± 4.46	6	Our group
2019	8	Hu Zhou	Selected sheep	50.73 ± 4.08	6	Our group
2023	12	Li Shui	common sheep	28.73 ± 4.78	6	Our group
2023	12	Li Shui	Selected sheep	37.23 ± 2.76	6	Our group
2024	10	Hang Zhou	Selected sheep	65.31 ± 4.43	8	Our group

**Table 3 animals-14-03047-t003:** Changes in lambing rate of Hu sheep.

Year	Sample Size	Area (Province)	Type	Slaughter Weight	Data Sources
1986	382	Zhe Jiang	purebred	214.1	Lv Baoquan [26]
1998	548	Hang Zhou	purebred	231.7	Xie Zhuang [27]
1999	1000	Jiang Su	purebred	232.2	Wang Yuanxing [28]
2006	118	Jiang Su	purebred	250.44	Wang Minghai [29]
2009	108	Xin Jiang	crossbred	193	Yang Yonglin [30]
2014	1000	He Nan	crossbred	221	Han Yufei [21]
2019	60	An Hui	purebred	262	Jiang Xichun [31]
2019	60	An Hui	crossbred	220	Jiang Xichun [31]
2022	30	He Bei	purebred	253.3	Zhang Tianhao [32]
2023	1253	Xin Jiang	crossbred	227.4	Mao Mingwei [33]
2023	120	Zhe Jiang	purebred	240.9	Our group

## Data Availability

The original contributions presented in this study are included in the article; further inquiries can be directed to the corresponding author.

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
