# Peer review of "How Food Consumption Trends Change the Direction of Sheep Breeding in China"

_animals, 2024, doi:10.3390/ani14213047_

Round 1

Reviewer 1 Report

Comments and Suggestions for Authors

I carefully analyzed the article numbered 3151476 in the journal Animals, entitled "How Food Consumption Trends Change the Direction of Sheep and Goat Breeding" . This paper primarily examines indicators such as the production and consumption of sheep and goat meat in China since 2010, with a specific focus on the Hu sheep to introduce the biological characteristics and related research progress of major representative breeds in Southern China. Overall, this paper is worthy of publication and introducing the development trends and characteristics of the sheep and goat industry in China to scientists worldwide. However, there are several issues that need to be addressed before it can be accepted for publication:

1. The paper contains some noticeable Chinglish, which should be polished and corrected. For instance, "Dupo sheep" is the Chinese transliteration, which I presume should be "Dorper sheep"; "Small-tailed Cold Sheep" should be "Small-tailed Han Sheep". In Table 1, there should be a space between each indicator and its unit in English format, for example, "price inde(%)" is incorrect and should be "price index (%)". "In the 12th century" should be corrected to "In the 12th century," with the "th" superscripted. There are many other minor issues like these, but I won't list them all here.

2. L99-103: "a formula adopted by Wang Jating [4] was used to calculate the self-sufficiency rate (SSR) of livestock and poultry products. SSR=P/(P+I-E-S)*100%." This is a very key indicator in the methodology of this paper. It should be emphasized why this indicator was chosen for evaluation.

3. The writing format of the affiliations needs to be changed: The "College of Animal Science and Technology" and "College of Animal Medicine" both belong to Zhejiang A&F University, so these two colleges should not be presented in parallel.

4. In Figure 1, four indicators are represented by only two colors, which can easily cause confusion. It is recommended to use four different colors to make it more intuitive and clear.

5. What I find most difficult to understand is the content in Figure 1 and Table 1. When indicating the years, why only go up to 2021 (or at most 2022)? We are already well into 2024, and the data for 2023 should have been published or reported. Why weren't these data adopted and analyzed up to 2023? These data are crucial for trend analysis, but they are not used in this paper. Please clarify this, as it affects my judgment.

6. The headers for Table 1 and Table 3 should be above the table, not below.

7. In Figure 3, image A should be taken in the aisle of the pen to fully show the situation on both sides. For image B, please replace it with a cleaner one, as this image does not demonstrate the characteristics of Hu sheep farming in China.

8. The citation format for references in the Table is incorrect.

9. I am not sure if the English representation of each province of China in Figure 2 is correct; please verify this. Additionally, the size of the Chinese map should include a scale. 

10. There are some issues in the Acknowledgments section; please delete it.

11. Due to the lack of data from Taiwan Province, "China" in the text should be changed to "China (Mainland).

12. L274-289: There are two large paragraphs here with several hundred words, but not a single reference. Additionally, the statement 'There are 104 local sheep and goat breeds in China, including 44 sheep and 60 goats' must be supported by a reference  at least. Furthermore, this very key data needs to be verified.

Finally, I would like to emphasize that the 12 points I mentioned above may seem overly critical, but for someone aiming to publish an international paper, they must be meticulous and rigorous.

Comments on the Quality of English Language

I carefully analyzed the article numbered 3151476 in the journal Animals, entitled "How Food Consumption Trends Change the Direction of Sheep and Goat Breeding" . This paper primarily examines indicators such as the production and consumption of sheep and goat meat in China since 2010, with a specific focus on the Hu sheep to introduce the biological characteristics and related research progress of major representative breeds in Southern China. Overall, this paper is worthy of publication and introducing the development trends and characteristics of the sheep and goat industry in China to scientists worldwide. However, there are several issues that need to be addressed before it can be accepted for publication:

1. The paper contains some noticeable Chinglish, which should be polished and corrected. For instance, "Dupo sheep" is the Chinese transliteration, which I presume should be "Dorper sheep"; "Small-tailed Cold Sheep" should be "Small-tailed Han Sheep". In Table 1, there should be a space between each indicator and its unit in English format, for example, "price inde(%)" is incorrect and should be "price index (%)". "In the 12th century" should be corrected to "In the 12th century," with the "th" superscripted. There are many other minor issues like these, but I won't list them all here.

2. L99-103: "a formula adopted by Wang Jating [4] was used to calculate the self-sufficiency rate (SSR) of livestock and poultry products. SSR=P/(P+I-E-S)*100%." This is a very key indicator in the methodology of this paper. It should be emphasized why this indicator was chosen for evaluation.

3. The writing format of the affiliations needs to be changed: The "College of Animal Science and Technology" and "College of Animal Medicine" both belong to Zhejiang A&F University, so these two colleges should not be presented in parallel.

4. In Figure 1, four indicators are represented by only two colors, which can easily cause confusion. It is recommended to use four different colors to make it more intuitive and clear.

5. What I find most difficult to understand is the content in Figure 1 and Table 1. When indicating the years, why only go up to 2021 (or at most 2022)? We are already well into 2024, and the data for 2023 should have been published or reported. Why weren't these data adopted and analyzed up to 2023? These data are crucial for trend analysis, but they are not used in this paper. Please clarify this, as it affects my judgment.

6. The headers for Table 1 and Table 3 should be above the table, not below.

7. In Figure 3, image A should be taken in the aisle of the pen to fully show the situation on both sides. For image B, please replace it with a cleaner one, as this image does not demonstrate the characteristics of Hu sheep farming in China.

8. The citation format for references in the Table is incorrect.

9. I am not sure if the English representation of each province of China in Figure 2 is correct; please verify this. Additionally, the size of the Chinese map should include a scale. 

10. There are some issues in the Acknowledgments section; please delete it.

11. Due to the lack of data from Taiwan Province, "China" in the text should be changed to "China (Mainland).

12. L274-289: There are two large paragraphs here with several hundred words, but not a single reference. Additionally, the statement 'There are 104 local sheep and goat breeds in China, including 44 sheep and 60 goats' must be supported by a reference  at least. Furthermore, this very key data needs to be verified.

Finally, I would like to emphasize that the 12 points I mentioned above may seem overly critical, but for someone aiming to publish an international paper, they must be meticulous and rigorous.

Reviewer 2 Report

Comments and Suggestions for Authors

Manuscript ID: animals-3151476 

Type of manuscript: Review
Title: How Food Consumption Trends Change The Direction of Sheep and Goat
Breeding
Authors: Wang Xiaoyu, Shen Wei, Wu Pan, Wang Chengli, Li Jiahua, Wang Di, Yue
Wanfu *

The authors discuss how food consumption trends have influenced the direction of sheep and goat breeding, particularly the shift in sheep and goat farming and the development of the Hu sheep industry in China. The article reflects the improvement of Chinese people's consumption ability and the change of consumption direction from the change of sheep breeding direction, providing China's development experience for today's lagging regions of the global livestock industry. Although this work is useful, several concerns were raised throughout the text and need to be clarified by the authors before further consideration. See comments below:

L8: Consider rephrasing "global consumer market" to "global market" or "global consumer demand" for clarity.

L33: Add a sentence or two to summarize the key findings and implications.

L44: What specific developments in the cotton and hemp industries led to the decline of the wool market? Could you provide more details or examples?

L49-50: Why are Dupo sheep, Boer goats, and Australian white sheep mentioned specifically? What makes these breeds significant compared to others?

L77-81: The sentence structure is a bit complex. Consider breaking it down into two or three shorter sentences for better readability.

Figure 1: Data should be updated to the year 2023 or 2024. Check all.

Page 3: Add the title of the table.

L124: The term "traditional meat-producing breeds" might be unfamiliar to some readers. Consider providing a more specific definition or example.

Figure 2: Consider adding a legend to the maps to clarify the meaning of the different colors or shades.

L204: The term "barn feeding" might be unfamiliar to some readers. Consider providing a more specific definition or alternative terminology.

L252: "feed conversion rate"?-àLine 252: "feed conversion ratio?

L366: Please define “conventional nutrients”??

L404-409 et al: Too long sentence and hard to understand. Please separate it into 2-3 sentences. Check throughout manuscript.

Page 12: Add title of table.

Lines 474: Provide more details about the "project of 'retreating from pasture to grass.'"

L518-519: Please clarify "highly organized, advanced in concept"

L533-557: Please clarify whether this conclusion is aligned with the title or objective yet. Please revise.

Comments on the Quality of English Language

Extensive editing of English language required.

Reviewer 3 Report

Comments and Suggestions for Authors

Simple summary: nice and simple, no change necessary.

Abstract: No changes needed.

1. Sheep and goat farming in the consumer shift: Well described. Figure 1 legend will be better if authors add data source.

2. Transformation of China's sheep and goat farming industry: no comments. In table 1, please mention data source in table legend. Please check this for all figures and tables in the manuscript.

3. Several Direction Shifts in Chinese Sheep Breeding: This portions was bit too long.,3.1, and 3.2 can be concise. 

4. Breeding achievements of Hu sheep for meat use: Well written but section 4.3 is too long.

5. Conclusions: no comments.

Round 2

Reviewer 1 Report

Comments and Suggestions for Authors

The revised manuscript meet my all concern and smooth writting. I agree it to  publish in the anminals journal.

Comments on the Quality of English Language

smooth writing after revision

Author Response

Thank you very much for your guidance on this review. Your advice has had a very positive impact on my writing skills while improving this article. It is my honor to have been taught and recognized by you.

Reviewer 2 Report

Comments and Suggestions for Authors

The manuscript was improved and no further suggestions.

Author Response

(The authors gave the same response as above.)
